# Does Urban Forest Control Smog Pollution? Evidence from National Forest City Project in China

**Hanjin Xie [1], Xi Tan [1,*], Chunmei Yang [1] and Cheng Li [2]**

1  School of Economic and Management, East China Jiaotong University, Nanchang 330013, China
2  School of Economics, Zhongnan University of Economics and Laws, Wuhan 430000, China
*  Correspondence: 2020048025100008@ecjtu.edu.cn

**Abstract:** The National Forest City (NFC) project is an important measure to promote the urban environment in China, but its environmental performances have not been fully evaluated yet. This paper uses difference-in-differences (DID) to evaluate the smog pollution controlling effects and mechanisms of the NFC project based on the panel data of 283 cities in China from 2000 to 2018. This study found the following: (1) The NFC project significantly reduced smog pollution by 3.4% on average; the effect strengthened over time and rose to 8.5% in the 10th year after the NFC project. The average treatment effect was also confirmed by a series of robustness tests. (2) The NFC project can control smog pollution by greening urban space and greening social culture. (3) The treatment effect was related to both natural factors and human factors. The reduction in smog pollution was much stronger in the southern, hilly, warm and humid regions. Public willingness and government attention to environmental protection help with the smog pollution controlling of the NFC project as well.

**Keywords:** national forest city; urban forests; smog pollution; difference-in-differences; environment governance





## 1. Introduction

Smog pollution has become one of the most serious environment problems along with the booming of China's economy. It led to high economic cost and loss of public welfare. In order to promote land greening and urban environment, the Chinese government launched the National Forest City (NFC) project. From 2004 to 2019, the State Forestry and Grassland Administration of China successively issued a series of documents, including the industry standard <National Forest City Evaluation Index (LY/T2004—2012)>, the national standard <National Forest City Evaluation Index (GB/T37342-2019)> and <Approval Measures for the Title of National Forest City>. With the maturing of management systems, the NFC project, which provided a channel to increase urban greening and improve the economic sustainability, has been seen as being of great importance by Chinese local governments. Referring to <The Communique on China's Land Greening Status in 2020>, the number of cities intending to implement the NFC project has reached 441 by 2020. However, the effects of the NFC project on environment, economy and culture have not been valued, empirical evidence about that was still lacking.

Existing studies have shown that forest has a series of ecological functions such as climate regulation, carbon balance, soil and water conservation [1,2]. In particular, forest serves as a natural air filter, and its role in preventing smog pollution has been widely verified [3,4]. Additionally, as Figure 1 shows, the number of cities with forests increased rapidly from 2004 to 2018, correspondingly, the $PM_{2.5}$ concentration, by 2018, has dropped 38.65% from the highest point in 2011. These trends also reveal the potential causality of the NFC project and the improvement of smog pollution. So, our question is whether the NFC project controlled smog pollution? To answer this, taking the anti-smog function of

urban forest as a starting point, this paper creatively uses the natural experiment created by the NFC project to estimate the smog control effect of urban forest. Furthermore, we probe deeply into the mechanisms of the NFC project on smog pollution controlling besides the contribution of urban forest construction.

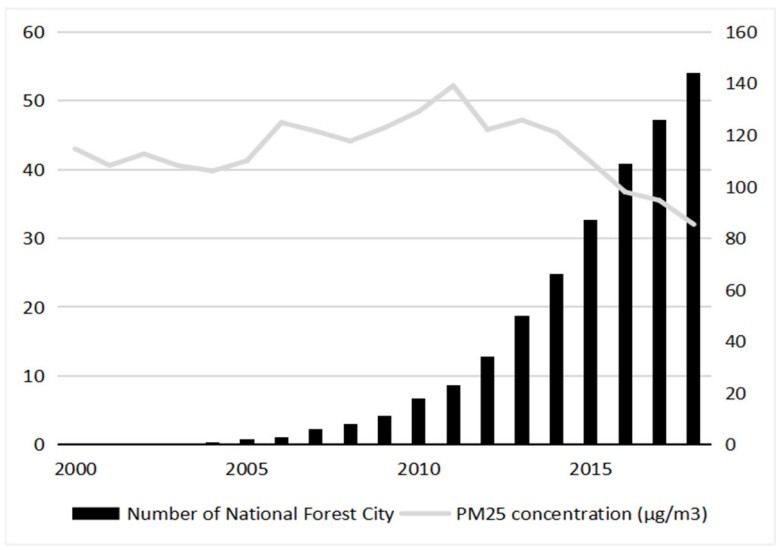

**Figure 1.** Trends in number of NFC project and $PM_{2.5}$ concentration from 2000 to 2018.

Compared to existing studies, this paper makes three main marginal contributions. Firstly, based on the natural experiment of the NFC project, we examined the relationship between smog pollution and urban forest construction to provide important empirical evidence about the environmental effects of urban forest. Secondly, by examining the mechanisms, it is found that the NFC project can not only green the urban space but also green the public's environmental behavior, proving that government initiatives to protect environment help to build greening social culture. Finally, heterogeneity analysis based on natural factors and human factors reveals the significance of differentiated urban forest construction and ecological education in environmental governance.

## 2. Literature Review

The concept of a forest city is about the organic integration of urban space and natural ecology; it is essentially about urban planning. So, what urban planning can be conducive to shape an environmentally friendly city? Based on the correlation between urban spatial structure and human activities, the scale and structure of urban space are related to environmental quality. To be specific, low-density urban space planning will lead to unacceptable consequences ranging from separation between jobs and housing locations, destruction of greenbelt and excessive housing construction; contrarily, compact and intensive urban space is conducive to emission reduction [5–8]. In addition to regulating human activities, environmentally friendly urban planning should also take environmental carrying capacity (*EEC*), the threshold which the urban ecological system is able to endure human activities, into consideration [9]. However, the carrying capacity of the urban environment is definite in a definite period and area. Pollutants cannot be completely degraded by the urban ecological system when the quantity of pollutants is beyond the environmental carrying capacity, resulting in a residue of pollutants [10]. Thus, with the promotion of environmental carrying capacity, there should be a substantial improvement of environment theoretically.

Basically, the environmental effect of a forest city is rooted in the ecological function of urban forest. Researches show that urban forest is capable of improving air quality. Chen et al. [11], taking large cities in China as study objects, found that urban green space can significantly suppress smog pollution. Matos et al. [12] proved the negative correlation between air quality and green area in 42 urban green spaces in Lisbon, Portugal. It was

also found that the larger the vegetation density, the better the air quality. The function of forest to absorb and degrade air pollutants is the key for urban green space to suppress smog pollution. Existing studies indicate that the forest belt can block the air flow and thus result in the sedimentation of air particles; additionally, the tree canopy can reduce dust and avoid secondary pollution by covering the ground, and the blade surface has the function of attaching to particles as well [13–15]. Furthermore, plants can degrade harmful components in gas or store them in organs by exchanging gas with outside through stomata and lenticels [16,17].

The research above has clarified the necessity of expanding environmental carrying capacity. They also provided the indirect evidences that forest cities may control smog pollution. However, the direct evidence on the relationship between forest city and smog pollution is deficient; conclusions so far mainly originate from the assumption that urban forests absorb and degrade pollutants, while examination based on empirical data is still lacking.

## 3. Theoretical Analysis

### 3.1. Background of NFC Project

In 2004, the first China Urban Forest Forum awarded Guiyang the title of "National Forest City". Since then, the initiative to construct a forest city has become nationwide. The primary objective of the NFC project was to set an example of urban greening, while the NFC project has gradually become an important platform for cities to promote ecological construction as the practice deepens. In 2007, the <National Forest City Evaluation Index> issued by the State Forestry and Grassland Administration put forward seven targets of the NFC project, including forest ecosystems, green rate, ecological culture, etc. It also set up a series of rules and regulations, such as: Construction funds are included in local government budget; The planning of the NFC project shall be incorporated into the urban master planning; Adopting a reassessment once every three years. In 2012, the industry standard <National Forest City Evaluation Index (LY/T2004—2012)> further refined the evaluation system. Specifically, the new standard proposed a joint investment mechanism of government social strength. In 2019, the national standard <National Forest City Evaluation Index (GB/T37342-2019)> stipulates that the construction period in the planning of the NFC project shall be more than 10 years. The NFC project was first written into the <Law of the People's Republic of China on Forest>. With the maturing of institution, the NFC project has become an important channel to green urban space and social culture, and it has indeed made great contributions to ecological civilization.

### 3.2. Theoretical Hypothesis

#### 3.2.1. NFC Project and Smog Pollution Control

Environment carrying capacity reflects the support of an urban ecosystem for economic development [18]. Therefore, the preservation of environment should focus on the environment carrying capacity. Urban forest can absorb and degrade air pollutants; thus, it is an irreplaceable component of atmospheric environment carrying capacity supply. The institutional arrangements of the NFC project can ensure the accomplishment of urban forest construction; consequently, urban atmospheric environment carrying capacity should be effectively improved, that is, the NFC project helps to control smog pollution.

Natural factors and human factors may affect the smog control achievement of the NFC project. In terms of natural factors, on the one hand, climate conditions, such as precipitation and temperature, can influence the formation of urban forest ecosystems [19,20]. On the other hand, the function of plants that are absorbing and degrading air pollutants is also determined by the plant species and vegetation characteristics [21], which may vary with geographical distribution.

Meanwhile, the willingness of public and local government, the two main participants in the project of NFC, to invest in environmental protection may also affect the smog control achievement. To the public, the stronger the willingness to invest in environmental

protection, the easier it is to accept the ecological publicity and education. Consequently, it is more likely for people to adopt a green lifestyle. For the local government, apparently, the smog control achievement of the NFC project is closely correlated to the attention that local governments paid to environmental governance. Specifically, local governments are more likely to mobilize resources and stick strictly to the planning of the NFC project.

Thus, hypothesis 1 and 2 were proposed:

**H1.** *Smog pollution shall be controlled after the NFC project.*

**H2.** *Natural factors and human factors may have a heterogeneity effect in the smog control achievement of the NFC project.*

### 3.2.2. Urban Space Greening

Urban forest construction is the core of the NFC project. To fulfil the planning of the NFC project on schedule and win the title of National Forest City, the local governments will invest more in urban forest construction and encourage social strength in greening, thus increasing the area of urban green space under the institution of the NFC project. Meanwhile, the NFC project makes sure that local governments improve the priority of urban forest construction in the urban master planning, so that the land use of green space can be allocated first. Consequently, the structure of urban land use will become more environmentally friendly, and the environmental pressure caused by the increasing industrial lands, commercial lands and residential lands can be alleviated as well. With the greening of urban space, the function of the overall green ecosystem related to absorbing and degrading pollutants should be strengthened [11,12].

Thus, hypothesis 3 was proposed:

**H3.** *The NFC project may control the smog pollution by greening urban space.*

### 3.2.3. Social Culture Greening

The project of NFC also aims at fostering the public's participation in afforestation and environmental protection under the guidance of local governments. 'Ecological culture' was arranged as one of the most important targets of the NFC project. Specifically, local governments were required to reach the pass line in the six aspects of 'ecological science popularization education spots', 'voluntary forestation', 'ecological science popularization education activity', 'ancient and famous trees protection', 'urban green plants protection' and 'public attitude to NFC project'. Then, people's willingness and cognition to environmental protection should grow by receiving ecological education and joining in ecological protection; thus, people would be more likely to adopt environmental responsible behaviors [22]. Improvement of living environment and city image will also help to enhance the people's happiness, self-identity with the city and satisfaction with the government, so that people are more inclined to adopt environmentally friendly behavior [23–25].

Thus, hypothesis 4 was proposed:

**H4.** *The NFC project may control the smog pollution by greening social culture.*

## 4. Model and Data

### *4.1. Baseline Model*

In this paper, the difference-in-difference (DID) model is used to examine the smog control effect of the NFC project on smog pollution. Within the framework of two-way fixed effect, the DID model, as shown in Formula (1), could estimate the treatment effect of the NFC project on smog pollution.

$$SP_{it} = \alpha + \beta NFC_{it} + X_{it}'\varphi + \mu_i + \gamma_t + \varepsilon_{it} \qquad (1)$$

where $i$ denotes the city and $t$ denotes the year. $SP_{it}$ denotes the smog pollution of city $i$ in year $t$. $NFC_{it}$ is the treatment variable and equals 1 for every year after the year that city $i$

start the NFC project. *X* denotes a group of control variables. $\mu$ denotes city fixed effects, $\gamma$ denotes year fixed effects.

### 4.2. Data Source

4.2.1. NFC Project

The sample in this study includes 283 cities from 2000 to 2018. Excluding county-level cities and cities with serious data loss, the number of cities that won the title of National Forest City by 2018 is 144, and the number of cities that won the title in 2019 but remained in the process of construction in 2018 is 19; thus, there are 163 cities treated and 120 untreated cities in the sample. Considering that the construction period of most cities is about 2 to 4 years, we set 3 years as the construction period for cities whose NFC project schedule was unavailable.

4.2.2. Smog Pollution

The $PM_{2.5}$ raster data released by the Atmospheric Composition Analysis Group (ACAG) [26] were selected as the measurement of smog pollution (logarithm), and the $PM_{2.5}$ data used in this study were extracted through ArcGIS software. The ACAG estimate ground-level fine particulate matter ($PM_{2.5}$) total and compositional mass concentrations by combining Aerosol Optical Depth (AOD) retrievals from the NASA MODIS, MISR, and SeaWIFS instruments with the GEOS-Chem chemical transport model, and they subsequently calibrated to regional ground-based observations of both total and compositional mass using Geographically Weighted Regression (GWR). Compared with AQI (Air Quality Index, ground-based monitor data from China), the $PM_{2.5}$ raster data have the longer time dimension to match up the history of the NFC project (2004-) and have wider geographical coverage (all cities in China) to avoid the data loss problem

4.2.3. Control Variables

The control variables in this study consist of economic variables and climate variables. In the baseline regression model, economic variables include *Pop* (total population at the end of the year), *Pgdp* (per capita income), *Pgdp2* (square of per capita income), *Is* (proportion of secondary and tertiary industry outputs), *Kg* (proportion of fixed asset investment), *Fdig* (proportion of foreign direct investment), *Pgas* (per capita gas consumption) and *Pbus* (per capita bus passenger transport times). The original data of economic variables come from the <China City Statistical Yearbook>. Climate variables include *Atem* (average temperature), *Aws* (average wind speed), *Apre* (accumulated precipitation), *Asun* (accumulated sunshine) and *Ahum* (average humidity). Those variables are calculated by daily data which originated from the China Meterological Administration. Descriptive statistics of above variables are shown in Table 1.

**Table 1.** Variables' Descriptive Statistics.

|  | **Obs** | **Mean** | **Sd** |
|---|---|---|---|
| Smog pollution | 5377 | 42.85 | 19.02 |
| NFC | 5377 | 0.219 | 0.414 |
| Economic variables |  |  |  |
| *Pop* | 5377 | 427.2 | 303.6 |
| *Pgdp* | 5377 | 2.390 | 3.141 |
| *Pgdp2* | 5377 | 0.156 | 0.783 |
| *Is* | 5377 | 0.842 | 0.099 |
| *Kg* | 5377 | 0.566 | 0.303 |
| *Fdig* | 5377 | 0.022 | 0.035 |
| *Pgas* | 5377 | 12.09 | 32.16 |
| *Pbus* | 5377 | 41.38 | 77.99 |

**Table 1.** *Cont.*

|  | **Obs** | **Mean** | **Sd** |
|---|---|---|---|
| Climate variables |  |  |  |
| *Atem* | 5377 | 14.46 | 5.155 |
| *Aws* | 5377 | 2.147 | 0.533 |
| *Apre* | 5377 | 1014,2 | 5364 |
| *Asun* | 5377 | 1996 | 491.4 |
| *Ahum* | 5377 | 0.693 | 0.092 |

## 5. Empirical Results

### 5.1. Impact of NFC Project on Smog Pollution

　　In the baseline regressions, fixed effects, economic variables and climate variables are controlled in order (Table 2, columns 1–3). The results suggest a significantly negative correlation between *SP* and *NFC* at 1%. It indicates that the smog pollution has been through an improvement after the NFC project. On average, the NFC project can reduce the $PM_{2.5}$ concentration by 3.4% (Table 2, columns 3). However, considering the construction period of the NFC project and the formation process of a forest ecosystem, the average treatment effect may not fully reflect the environmental performance of the NFC project, so it is necessary to further estimate the smog control effect of the NFC project in each period. Therefore, the annual treatment effects will be examined in the Section 5.3.

**Table 2.** The Impact of NFC Project on Smog Pollution.

|  | (1) | (2) | (3) | (4) | (5) | (6) |
|---|---|---|---|---|---|---|
|  | TW-FE | TW-FE | TW-FE | IV | IV | IV |
| *NFC* | −0.038 *** | −0.038 *** | −0.034 *** | −0.158 *** | −0.113 *** | −0.111 *** |
|  | (−2.62) | (−2.96) | (−2.70) | (−5.78) | (−4.36) | (−4.22) |
| *Pop* |  | 0.018 | −0.009 |  | 0.091 | 0.0667464 |
|  |  | (0.10) | (−0.05) |  | (0.93) | (0.73) |
| *Pgdp* |  | 0.092 | 0.087 |  | 0.010 *** | 0.010 *** |
|  |  | (1.51) | (1.49) |  | (3.30) | (3.15) |
| *Pgdp2* |  | −0.039 ** | −0.036 ** |  | −0.001 *** | −0.001 *** |
|  |  | (−2.57) | (−2.43) |  | (−4.60) | (−4.41) |
| *Is* |  | −0.666 *** | −0.621 *** |  | −0.007 *** | −0.006 *** |
|  |  | (−4.47) | (−4.34) |  | (−7.23) | (−6.95) |
| *Kg* |  | 0.015 | 0.009 |  | 0.019 | 0.013 |
|  |  | (0.62) | (0.37) |  | (1.02) | (0.71) |
| *fdig* |  | 0.157 | 0.144 |  | 0.173 | 0.158 |
|  |  | (0.76) | (0.77) |  | (0.94) | (0.93) |
| *Pgas* |  | 0.050 | 0.007 |  | 0.033 | −0.009 |
|  |  | (0.27) | (0.04) |  | (0.24) | (−0.06) |
| *Pbus* |  | −0.004 | −0.001 |  | −0.005 | −0.002 |
|  |  | (−0.54) | (−0.18) |  | (−0.83) | (−0.37) |
| *Atem* |  |  | 0.032 *** |  |  | 0.034 *** |
|  |  |  | (2.82) |  |  | (3.72) |
| *Aws* |  |  | −0.073 *** |  |  | −0.072 *** |
|  |  |  | (−2.84) |  |  | (−4.31) |
| *Apre* |  |  | −0.006 *** |  |  | −0.006 *** |
|  |  |  | (−5.89) |  |  | (−5.01) |
| *Asun* |  |  | −0.268 *** |  |  | −0.257 *** |
|  |  |  | (−8.97) |  |  | (−10.97) |
| *Ahum* |  |  | −0.685 *** |  |  | −0.654 *** |
|  |  |  | (−4.87) |  |  | (−5.15) |
| *Awr* × *Pnfc* |  |  |  | 0.020 *** | 0.021 *** | 0.021 *** |
|  |  |  |  | (32.66) | (32.12) | (31.98) |

**Table 2.** *Cont.*

| | (1) | (2) | (3) | (4) | (5) | (6) |
|---|---|---|---|---|---|---|
| | TW-FE | TW-FE | TW-FE | IV | IV | IV |
| Cragg–Donald Wald F-statastic | | | | 759.101 | 767.205 | 759.68 |
| City fixed effects | YES | YES | YES | YES | YES | YES |
| Year fixed effects | YES | YES | YES | YES | YES | YES |
| Observations | 5377 | 3933 | 5377 | 3933 | 5377 | 3933 |
| R-squared | 0.3289 | 0.3027 | 0.3455 | 0.3235 | 0.3705 | 0.3505 |

Notes: Figures in parentheses are *t* values, **, *** denote statistical significance levels at 10%, 5% and 1% respectively.

### 5.2. Endogeneity

Although the DID model helps to alleviate endogeneity theoretically, there are still endogeneity problems left in the baseline regression. To be specific, the better the urban ecological foundation, the less difficult it is for local government to implement the NFC project; hence, it is more likely for local government to win the NFC title. Meanwhile, cities with better ecological foundation may have better air quality as well. Considering data availability and validity, it is nearly impossible to control the impact of ecological foundation in the baseline regression; thus, the treatment effect may be miscalculated. Therefore, the product of *Awr* (average water resources) and *Pnfc* (the number of cities that implement the NFC project in a province) was selected as an instrumental variable to conduct Heckman two-step regression [27].

The implementing of the NFC project relies on water resources, it is easy for cities that have abundant water resources to construct an urban forest; on the contrary, it is much tougher for arid cities. However, smog pollution is not so strongly correlated with water resources, which are mainly composed of river water, reservoir water and ground water. Unfortunately, the variable total water resources is short of time variation because it is unavailable before 2017. To address this], *Awr*, which is the average of total water resources in 2017 and 2018 was calculated firstly; then, we multiplied *Awr* by *Pnfc* to obtain the final instrumental variable *Awr* × *Pnfc*, where *Pnfc* is the number of cities that implemented the NFC project in a province. The variable *Pnfc* which indicates whether provincial governments attach importance to the NFC project is also strongly correlated with the explanatory variable NFC, while local smog pollution cannot be influenced by the NFC project of other cities in the same province theoretically.

The results of two-step regressions show that *NFC* is still significantly negative, which is consistent with the baseline regression (Table 2, columns 4–6, significant at 1%). The coefficients of IV were all significantly positive, and the Cragg–Donald Wald F-statistic for weak instrumental variable identification is higher than the critical value 16.38 at the 10% level, indicating that the instrumental variable selected is appropriate.

### 5.3. Pre-Treatment Trend Test

The difference-differences strategy should meet the parallel trend assumption, as the treatment effect may be driven by a pre-existing difference between the treated and untreated subclass. Therefore, referring to Alder et al. [28], we use an event study model which is shown in Formula (2) to check the pre-treatment trend.

$$PM_{2.5it} = \alpha + \sum_{k=-7}^{k=10} \beta_k \times DT_{ik} + X_{it}'\varphi + \mu_i + \gamma_t + \varepsilon_{it} \tag{2}$$

where $DT_{ik}$ is a group of dummy variables, and $DT_{ik}$ is equal to 1 for the *k*-th year of city i after treated; otherwise, $DT_{ik}$ is equal to 0. $\beta_k$ measures the differential changes in smog pollution between the treated and untreated subclass in the year *k* after the NFC project. Note that the sample period is shortened to 17 years which include seven years before

treated and ten years after treated. $\beta_k$ measures the differential changes in smog pollution between the treated and untreated subclass in the year $k$ after the NFC project.

The regression results of the pre-treatment trend test are reported by the line of $\beta_k$ (Figure 2, at 10%), which includes 16 periods (the 7th year before treated was used as baseline). Apparently, $\beta_k$ values where $k$ is less than 0 are not statistically significant, which reveals no systematic difference in the pre-treatment trend across the two subclasses.

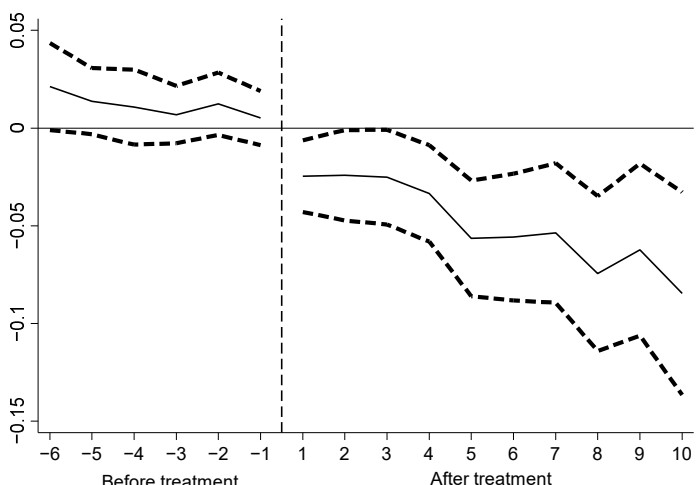

**Figure 2.** The differential changes in smog pollution: treated versus untreated subclasses.

### 5.4. Annual Treatment Effects

The regression result of the event study model also shows the annual treatment effects of the NFC project. Here, the $\beta_k$ values where k is greater than 0 are significantly negative, and the line shows a downward trend, which indicates that the annual treatment effects of the NFC project strengthen over time. Furthermore, to compare with the average treatment effect, the values of $\beta_k$ after the treatment were reported (Table 3). It shows that the treatment effect in the year 10 ($\beta_{10}$, 8.5%) is higher than the double average treatment effect (3.4%).

**Table 3.** Annual Treatment Effects.

|  | $DT_1$ | $DT_2$ | $DT_3$ | $DT_4$ | $DT_5$ |
|---|---|---|---|---|---|
| $\beta_k$ | −0.025 | −0.024 | −0.025 | −0.033 | −0.056 |
|  | $DT_6$ | $DT_7$ | $DT_8$ | $DT_9$ | $DT_{10}$ |
|  | −0.056 | −0.054 | −0.074 | −0.062 | −0.085 |

### 5.5. Placebo Test

To improve the reliability of the basic results in Section 5.1, we conducted a placebo test by sampling randomly 1000 times in the set of cities and the set of years. Specifically, first, every city was assigned one of the years between 2000 and 2018 as the treated year firstly; after that, a new treated subclass which includes 163 cities was randomly divided from the set of cities. Next, the baseline regression model was conducted by using the new treated subclass. Finally, the above steps were repeated 1000 times to obtain $\beta^{random}$ (the coefficient vector of the new treatment variables) and the corresponding *p*-values.

Here, the kernel density distribution of $\beta^{random}$ (Figure 3) is close to a normal distribution. The coefficient of *NFC* in baseline regression (−0.034, reference line) is located at the edge of distribution (cumulative probability lies in 1.1% to 1.3%), and the majority of *p*-values of $\beta^{random}$ are above 0.1 (Figure 4), which means most of the coefficients are not significant. Thus, the null hypothesis of the placebo test can be rejected.

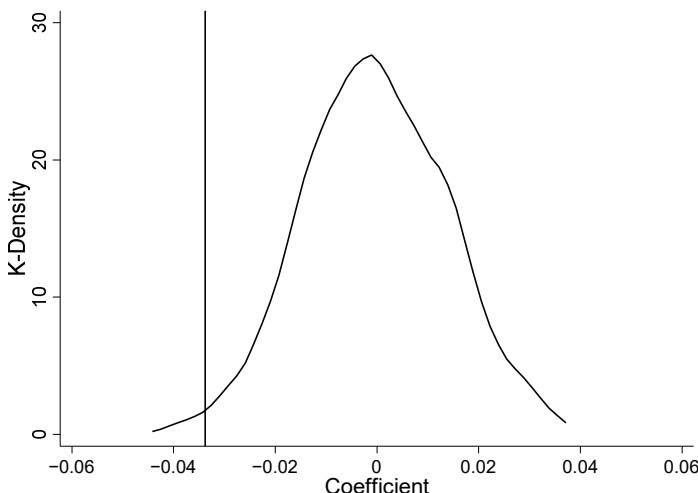

**Figure 3.** Kernel Density Distribution of $\beta^{random}$.

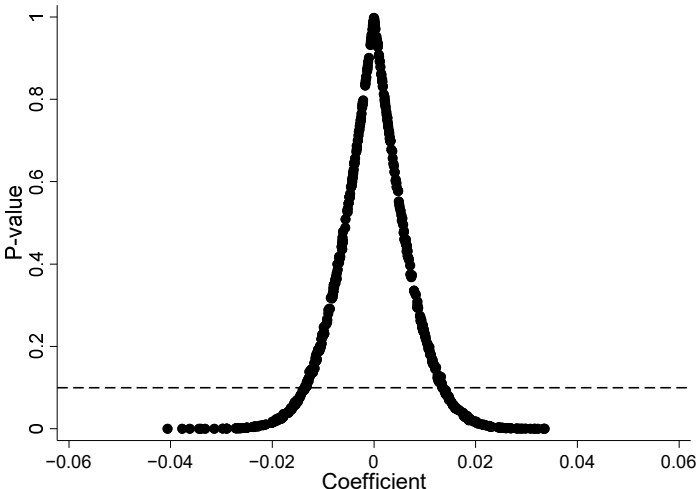

**Figure 4.** Scatter Distribution of P-value to $\beta^{random}$.

### 5.6. Pre-Existing Trend

Although the pre-treatment tread was examined in Section 5.3, line $\beta_k$ in Figure 3 represents a downward trend before the treatment (NFC project), which may indicate an overestimation in the baseline regression. Therefore, three types of pre-existing trend checks were conducted by controlling *Treat* $\times$ *T* (linear trend *T* of cities that treated), $\mu \times \gamma$ (interaction terms of city fixed effects and year fixed effects), and $L_\theta \times T$ (linear trend *T* of location factors $L_\theta$, $\theta$ ranges from 1 to 3). Here, $L_1$ is the dummy variable which is equal to 1 for every city belonging to the Special Economic Zone. $L_2$ is the dummy variable which is equal to 1 for every city located in the Beijing–Tianjin–Hebei region. $L_3$ denotes the minimum distance to Tianjin port, Shanghai port and Hong Kong port.

The results of pre-existing trend checks show the weaker significance and smaller coefficients of *NFC* compared with baseline regression (Table 4, columns 1–3). More importantly, it still suggests a significant correlation between NFC and smog pollution, which confirms the basic conclusion once again.

**Table 4.** Pre-Existing Trend Control.

| | (1) | (2) | (3) |
|---|---|---|---|
| *NFC* | −0.017 * <br> (−1.66) | −0.021 ** <br> (−2.23) | −0.026 ** <br> (−2.12) |
| *Treat* × *T* | −0.003 <br> (−1.35) | | |
| $L_1$ × *T* | | | −0.003 <br> (−1.32) |
| $L_2$ × *T* | | | 0.011 *** <br> (6.27) |
| $L_3$ × *T* | | | −0.025 *** <br> (−3.65) |
| $\mu$ × $\gamma$ | NO | YES | NO |
| Control variables | YES | YES | YES |
| City and Year fixed effects | YES | YES | YES |
| Observations | 5337 | 5337 | 5337 |
| R-squared | 0.3569 | 0.3721 | 0.3846 |

Notes: Figures in parentheses are *t* values, *, **, *** denote statistical significance levels at 10%, 5% and 1%, respectively.

### 5.7. Robust Test

This section represents a series of robustness tests, including data replacing, sample adjustment and controlling other environmental policies.

For the reason mentioned in Section 4.2.2, we use $PM_{2.5}$ raster data, while not AQI, as the measurement of smog pollution in the examinations above. However, in this part, the $PM_{2.5}$ raster data were replaced by AQI and the $PM_{2.5}$ and $PM_{10}$ data of the index to check whether the result depends on the measurement of explained variable. As AQI has a short history, there are only 835 observations left (167 cities, 5 year from 2014 to 2018) in the regression (Table 5, columns 1–3).

**Table 5.** Robustness Test.

| | (1) | (2) | (3) | (4) | (5) | (6) |
|---|---|---|---|---|---|---|
| | **AQI** | **$PM_{2.5}$** | **$PM_{10}$** | **Subclass 1 Excluded** | **Subclass 2 Excluded** | **Environmental Policies** |
| *NFC* | −7.111 *** <br> (−2.62) | −6.468 *** <br> (−2.72) | −6.622 ** <br> (−1.97) | −0.043 *** <br> (−3.36) | −0.052 *** <br> (−3.57) | −0.033 *** <br> (−2.65) |
| *Lcc* | | | | | | −0.015 <br> (−0.92) |
| *Kmc* | | | | | | 0.021 <br> (0.93) |
| *Sesap* | | | | | | −0.050 ** <br> (−2.11) |
| Control variables | YES | YES | YES | YES | YES | YES |
| City and Year fixed effects | YES | YES | YES | YES | YES | YES |
| Observations | 835 | 835 | 835 | 5301 | 4503 | 5337 |
| R-squared | 0.5714 | 0.7205 | 0.7138 | 0.3677 | 0.4115 | 0.3569 |

Notes: Figures in parentheses are *t* values, **, *** denote statistical significance levels at 10%, 5% and 1%, respectively.

In the second test, three kind of subclasses were excluded in the regression. The first subclass consists of Beijing, Tianjin, Shanghai and Chongqing: cities whose annual accumulated precipitation is less than 400 mm. Those cities were excluded, for their construction scope of the NFC project is different from that of other cities. The second subclass consists of cities that won the NFC title after 2016 and those that started implementing the NFC project before 2007; these were excluded for the potential overestimation or underestimation caused by the change of regulations and rules.

In the third test, three environmental policies, including Low Carbon City (Lcc), PM$_{2.5}$ Key Monitoring City (Kmc) and Special Emission Standards for Air Pollutants (Sesap), were controlled in the regression for their policy effects on smog pollution, which is basically the same as the NFC project's.

The results indicate the treatment variable NFC is statistically significant in the three robustness test all along (Table 5, columns 1–6, at 5%). Thus, the basic conclusion is still robust so far.

## 6. Mechanism and Heterogeneity

### 6.1. Mechanism

In this section, the mechanisms of the NFC project on smog pollution will be examined in the two aspects of urban space greening and social culture greening according to Section 3.2.

#### 6.1.1. Urban Space Greening

Land use greening, the primary mission of the NFC project, is an appropriate measurement of urban space greening. Specifically, the variables green area (Ga) and greening rate (Gr) that originated from the <China Urban Construction Statistical Yearbook> were selected as explained variables. The results show that the coefficients of *NFC* are both significantly positive (Table 6, columns 1–2), which means that the NFC project has indeed greened urban space statistically. Meanwhile, as proved in the existing studies, greener urban space is conducive to controlling smog pollution [11,12]. Thus, the mechanism that the NFC project can control smog pollution by greening urban space has been checked.

**Table 6.** Mechanism.

|  | (1) | (2) | (3) | (4) |
|---|---|---|---|---|
|  | **Ga** | **Gr** | **Tcc** | **Pcc** |
| *NFC* | 1.695 ** | 0.009 * | −0.084 ** | −0.077 ** |
|  | (2.07) | (1.80) | (−2.43) | (−2.16) |
| Control variables | YES | YES | YES | YES |
| City and Year fixed effects | YES | YES | YES | YES |
| Observations | 4981 | 4979 | 5377 | 5377 |
| R-squared | 0.2705 | 0.0860 | 0.7362 | 0.6892 |

Notes: Figures in parentheses are *t* values, *, ** denote statistical significance levels at 10%, 5% and 1%, respectively.

#### 6.1.2. Social Culture Greening

The greening of social culture is reflected in the greening of lifestyle. In the examination, the variables total consumption of electricity (*Tce*) and per capita consumption of electricity (*Pcc*) originated from the <China City Statistical Yearbook> were selected as explained variables. Apparently, the significant negative correlations between *NFC* and the two explained variables indicate that the energy consumption has been through a decline after the NFC project (Table 6, columns 3–4). That means from the perspective of resource-saving, people's lifestyle has greened. As the greening of lifestyle helps to improve smog pollution, the NFC project can control smog pollution by greening social culture.

### 6.2. Heterogeneity

#### 6.2.1. Heterogeneity of Natural Factors

The smog pollution control effect of the NFC project is rooted in the function of a forest ecosystem—absorbing and degrading pollutants—hence, the effect varies with forest types. Referring to existing study, the capacity of an evergreen broad-leaved forest to absorb air particles is significantly stronger than that of a deciduous broad-leaved forest [21]. Qinling-Huaihe, the north–south separatrix of China, divides its forest belt into two parts. In the south, evergreen broad-leaved is the main type of forest, while in the north, deciduous

broad-leaved forests dominate, so the treatment effect of the NFC project in the north of China may differ from that in the south as well.

Topography can influence the diffusion and accumulation of air pollutants. On the one hand, undulate topography can weaken the wind speed, resulting in the accumulation of air pollutants. On the other hand, undulate topography results in a larger area of the plants' contact to the air, which helps forest remove particles.

To check the heterogeneity of geography factors, cities were grouped by their geographical features. Specifically, cities were defined as a 'northern city' or 'southern city' according to cities' location relative to Qinling-Huaihe. Meanwhile, referring to Feng et al. [29], cities were defined as a 'plain city' if the city's relief degree of land surface (*rdls*) is less than 0.5 and were defined as a 'hilly city' if the city's relief amplitude is more than 0.5. Then, the treated subclass was divided into northern-hilly group, northern-plain group, southern-hilly group and southern-plain group with the control group kept the same. The regression results suggest a highly significant and negative net effect of the NFC project in the 'southern hilly city' and 'southern plain city' groups, but, by contrast, it is not significant in the 'northern hilly' and 'northern plain' groups. Meanwhile, the net effect in a 'hilly city' is higher than that in a 'plain city' in general (Table 7, columns 1–4).

**Table 7.** Natural Factors Heterogeneity.

| | (1) | (2) | (3) | (4) | (5) | (6) | (7) |
|---|---|---|---|---|---|---|---|
| | Northern Plain | Southern Plain | Northern Hilly | Southern Hilly | Atem | Apre | Ahum |
| *NFC* | 0.019 (0.87) | −0.053 *** (−3.72) | −0.014 (−0.46) | −0.118 *** (−5.81) | | | |
| *NFC × moderator* | | | | | −0.014 *** (−5.07) | −0.012 *** (−5.56) | −0.006 *** (−3.86) |
| *Atem/Apre/Ahum* | | | | | 0.039 *** (3.40) | −0.006 *** (−5.20) | −0.446 *** (−2.94) |
| Control variables | YES | YES | YES | YES | YES | YES | YES |
| City and Year fixed effects | YES | YES | YES | YES | YES | YES | YES |
| Observations | 2983 | 3420 | 2717 | 3097 | 5337 | 5337 | 5337 |
| R-squared | 0.3601 | 0.3574 | 0.3052 | 0.3530 | 0.3809 | 0.3795 | 0.3773 |

Notes: Figures in parentheses are *t* values, **, *** denote statistical significance levels at 10%, 5% and 1%, respectively.

Climate not only has a direct impact on smog pollution [30,31] but also has an indirect effect on air quality by affecting plant growth. To check the heterogeneity of climate factors, *Atem*, *Apre* and *Ahum* are selected as moderators in the regression model (Formula (3)), where $M_{\phi,it}$, $\phi$ ranges from 1 to 3, denoting the moderators *Atem*, *Apre* and *Ahum*, respectively. The regression results show that coefficients of the interaction terms are significantly negative (Table 7, columns 5–7), indicating that the warmer and the humid the climate, the stronger the smog control effect of the NFC project.

$$PM_{2.5it} = \alpha + \beta_1 Treat_{it} + \beta_2 Treat_{it} \times M_{\phi,it} + M_{\phi,it} + X_{it}'\varphi + \mu_i + \gamma_t + \varepsilon_{it} \tag{3}$$

### 6.2.2. Heterogeneity of Human Factors

The heterogeneity of human factors is also examined from the two aspects of public willingness to support environmental protection and government attention to environmental protection. Specifically, environmental petition per capita (*Pep*) is selected as the measurement of public willingness to support environmental protection, and environmental word frequency (*Ewf*) is selected as the measurement of government attention to environmental protection. Then, according to the median of the two grouping variables *Ewf* and *Pep*, the treated subclasses are grouped into high level and low level with the control group kept the same. Note that the original data of Pep comes from the <China Environment Yearbook>; *Ewf* is obtained by analyzing the word frequency of the local government annual work report. Here, the words used in the word frequency analysis

include low carbon, environmental protection, air, green, $PM_{2.5}$, COD, carbon dioxide, $PM_{10}$, ecology, emission, emission reduction, sulfur dioxide, and energy consumption.

The grouped regression results show that the coefficients of NFC are significantly negative while they are much higher in the high-level group (Table 8, columns 1–4). It indicates that, on the one hand, public willingness to environmental protection and government attention to environmental protection do affect the smog control effect of the NFC project, and the smog control effect of the NFC project always exists regardless of human will, as the function of urban forests to improve the air quality is unchangeable on the other hand.

**Table 8.** Human Factors Heterogeneity.

|  | (1) | (2) | (3) | (4) |
|---|---|---|---|---|
|  | High-Level of Pep | Low-Level of Pep | High-Level of Ewf | Low-Level of Ewf |
| *NFC* | −0.048 *** | −0.028 ** | −0.055 *** | −0.023 ** |
|  | (−4.59) | (−2.36) | (−5.16) | (−2.00) |
| Control variables | YES | YES | YES | YES |
| City and Year fixed effects | YES | YES | YES | YES |
| Observations | 3933 | 3724 | 3952 | 3705 |
| R-squared | 0.3569 | 0.3406 | 0.3518 | 0.3476 |

Notes: Figures in parentheses are *t* values, **, *** denote statistical significance levels at 10%, 5% and 1%, respectively.

## 7. Conclusions

This paper has used the NFC project in China as a natural experiment and taken the function of urban forest, absorbing and degrading air pollutants, as a starting point to examine whether urban forest can control smog pollution. The results of baseline regression indicate that the NFC project has a significant negative effect on smog pollution, which was 3.4% on average. With the parallel trend assumption verified, the basic result is robust in the placebo test, controlling for pre-existing trends. The basic result is confirmed by further tests, including data replacing, sample adjustment, and controlling for other environmental policies as well. Using the event study method, we have also examined the annual treatment effects which rose to 8.5% in the tenth year of the NFC project from 2.5% at the beginning.

Examinations of mechanisms were creatively conducted from the perspective of urban space greening and social culture greening. It was found that the NFC project has effectively promoted the greening of land use and public lifestyle, which were proved to benefit pollution regulation. Based on the results above, the smog control heterogeneity of natural factors and human factors were tested. Specifically, tests for heterogeneity of natural factors indicated that treatment effects in southern hilly and southern plain cities are much stronger than those in northern hilly and northern plain cities. Additionally, the warmer and the more humid the climate, the stronger the treatment effect of the NFC project. Test for heterogeneity of human factors revealed that encouraging public willingness to support environmental protection and the more attention to environmental protection that the government paid can maximize the environmental performance of the NFC project as well.

Although this study might be short of estimating the overall welfare effects of the NFC project, it provided empirical evidences for the environmental performance of urban forest and pointed out the importance of natural factors in urban land greening. It also suggests that urban forest may help with realizing a long-run conservation of air quality, which should be the focus of environment regulation in the future. Additionally, further analysis on mechanisms and heterogeneity reveals that public willingness to support environment protection and people's lifestyle can be the support for environment improvement; thus, publicity and education of environmental protection need to be valued more than ever.

**Author Contributions:** All authors contributed to the study conception and design. Material preparation, data collection and analysis were performed by H.X. and X.T. The first draft of the manuscript

was written by X.T., C.Y. and C.L., and all authors commented on previous versions of the manuscript. All authors have read and agreed to the published version of the manuscript.

**Funding:** This research was funded by National Natural Science Foundational of China, grant number 71864011.

**Institutional Review Board Statement:** Not applicable.

**Informed Consent Statement:** Not applicable.

**Data Availability Statement:** Data and materials are available from the authors upon request.

**Conflicts of Interest:** The authors declare no conflict of interest.

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
