# Peer review of "Does Urban Forest Control Smog Pollution? Evidence from National Forest City Project in China"

_sustainability, doi:10.3390/su141912897_

Round 1

Reviewer 1 Report

Dear Authors,

I have carefully reviewed the submission evaluating smog pollution controlling effects and mechanisms from National Forest City Project in China. The authors wrote the text brief and concise. The data were performed in appropriate statistics and presented with tables and graphs. But I found some minor points that need to be revised and indicated in the text. In my opinion, the ms should be worthy of publishing after minor revision.

Author Response

Dear Reviewer 1,

Thanks a lot for having reviewed our manuscript. Now we have revised the manuscript according to the reviewers’ comments. Most of the revisions are in the manuscript. Some explanations regarding the revisions of our manuscript are as follows.

Response: Now the methods in the abstract (line 20) are omitted and the missing sentence (line 463) have been deleted in the manuscript.

Reviewer 2 Report

Dear Authors,

Overall a good paper, but I would like to see the following addressed;

1) A lot of the analysis relies on the data provide by the Dalhousie Group (lines 192-193), but this is provided without;

a) a reference or web site

b) a statement on how it is produced, I am presuming it is something like remote sensing, but you don't say

c) a comparison with the ground based measurements mentioned from line 315.

2) why only PM2.5 when you have some data on PM10 (which may, or may not, be more easily captured by vegetation.

3) The reduction in smog appears to be statistically significant, but is it of practical significance? For example, is there any data on admissions to hospitals, that might indicate that the reduction is significant for health? Or that the levels fall below WHO standards after x years?

4) The title of Figure 1 is not very informative as the graph also includes post-treatment data.

5) As there appears to be a significant difference between the north and south is it also worth while to look at seasonal effects, that is reduction in winter smog, reduction in summer smog?

6) I would like to see some "scatter graphs", tables of figures are good, but I would like to be able to visualize how scattered the data are, and how the trends look.
